# Ethyl Acetate Fraction of *Punica granatum* and Its Galloyl-HHDP-Glucose Compound, Alone or in Combination with Fluconazole, Have Antifungal and Antivirulence Properties against *Candida* spp.

**DOI:** 10.3390/antibiotics11020265

**Published:** 2022-02-18

**Authors:** Aline Michelle Silva Mendonça, Cristina de Andrade Monteiro, Roberval Nascimento Moraes-Neto, Andrea Souza Monteiro, Renata Mondego-Oliveira, Camila Evangelista Carnib Nascimento, Luís Cláudio Nascimento da Silva, Lidio Gonçalves Lima-Neto, Rafael Cardoso Carvalho, Eduardo Martins de Sousa

**Affiliations:** 1Graduate Program in Microbial Biology, CEUMA University, UniCEUMA, São Luís 65055-310, MA, Brazil; alimichellemend@gmail.com (A.M.S.M.); andreasmont@gmail.com (A.S.M.); luiscn.silva@ceuma.br (L.C.N.d.S.); lidio.neto@ceuma.br (L.G.L.-N.); eduardo.martins@ceuma.br (E.M.d.S.); 2Biology Laboratory, Maranhão Federal Institute, IFMA, São Luís 65030-005, MA, Brazil; cristinamonteiro@ifma.edu.br; 3Graduate Program in Health Sciences, Federal University of Maranhão, UFMA, São Luís 65080-805, MA, Brazil; robervalmoraes11@gmail.com (R.N.M.-N.); camila.carnib@ufma.br (C.E.C.N.); 4Maurício de Nassau Faculty, UNINASSAU, São Luís 65040-840, MA, Brazil; re_mondego@hotmail.com; 5Graduate Program in Odontology, CEUMA University, UniCEUMA, São Luís 65075-120, MA, Brazil

**Keywords:** *Punica granatum*, galloyl-HHDP-glucose, in silico analysis, in vitro analysis, candidiasis

## Abstract

Candidiasis is the most common fungal infection among immunocompromised patients. Its treatment includes the use of antifungals, which poses limitations such as toxicity and fungal resistance. Plant-derived extracts, such as *Punica granatum*, have been reported to have antimicrobial activity, but their antifungal effects are still unknown. We aimed to evaluate the antifungal and antiviral potential of the ethyl acetate fraction of *P. granatum* (PgEA) and its isolated compound galloyl-hexahydroxydiphenoyl-glucose (G-HHDP-G) against *Candida* spp. In silico analyses predicted the biological activity of G-HHDP-G. The minimum inhibitory concentrations (MIC) of PgEA and G-HHDP-G, and their effects on biofilm formation, preformed biofilms, and phospholipase production were determined. In silico analysis showed that G-HHDP-G has antifungal and hepatoprotective effects. An in vitro assay confirmed the antifungal effects of PgEA and G-HHDP-G, with MIC in the ranges of 31.25–250 μg/mL and 31.25 ≥ 500 μg/mL, respectively. G-HHDP-G and PgEA synergistically worked with fluconazole against planktonic cells. The substances showed antibiofilm action, alone or in combination with fluconazole, and interfered with phospholipase production. The antifungal and antibiofilm actions of PgEA and G-HHDP-G, alone or in combination with fluconazole, in addition to their effects on reducing *Candida* phospholipase production, identify them as promising candidates for therapeutics.

## 1. Introduction

The incidence of fungal infections is increasing considerably in humans, due to the indiscriminate use of antibiotics, immunosuppressants, and increased use of invasive procedures, such as catheters. These infections can be debilitating, persistent, and result in costly treatments. Many of these microorganisms are natural colonizers of the human microbiota. However, they have an arsenal of factors and virulence properties that are associated with disorders in the host, such as immunodeficiency, trauma, and surgical procedures, which enable them to be opportunistic infections [1,2,3].

*Candida* spp. are commonly associated with diseases in humans, such as oral and vulvovaginal candidiasis, skin infections, and onychomycosis. These infections may occur even in immunocompetent hosts [4,5], when predisposing factors are present (age, antibiotic use, sexual activity, diabetes mellitus, and idiopathic causes), resulting in the possible need for prolonged treatments that may cause recurrences and generation of resistant species [6,7].

The main virulence factor of the *Candida* species, especially *C. albicans*, is its ability to form biofilms. Biofilms produced by pathogenic fungi are characterized by communities of filamentous fungi that adhere to biotic and abiotic surfaces, eventually expanding into highly organized communities that are resistant to antimicrobials and environmental conditions [8,9].

There are a few classes of antifungal agents available to treat *Candida* infections, but these have limitations in terms of their high cost and toxicity. Most infections caused by yeasts are preferably treated with Fluconazole (FCZ), an azole antifungal that acts by inhibiting ergosterol biosynthesis. FCZ has useful properties that make it the drug of choice, such as a wide spectrum of action, low cell toxicity compared to other antimycotics and high bioavailability. However, FCZ has a fungistatic action, inhibiting growth but not killing yeast cells, which can lead to the development of resistance [10].

Moreover, there is evidence of increased antifungal resistance to the available drugs, and most of them are poorly effective in treating diseases associated with biofilm formation [11,12,13,14]. Thus, there is a need to search for new antifungal compounds that are more effective, cheaper, and less toxic. Medicinal plants and their isolated compounds (e.g., cinnamaldehyde and eugenol) with antimicrobial properties are promising therapeutic alternatives for fungal infections [15].

*Punica granatum* is a plant belonging to the family Punicaceae. It is originally from Asia and is cultivated in several parts of the world, including Brazil [16]. Its fruits, roots, stems, and leaves are rich in tannins, flavonoids, ellagic acid, gallic acid, phenolic compounds, and other substances that have antioxidant, anti-inflammatory, antibacterial, and antifungal activities [17].

Our group previously characterized the ethyl acetate fraction obtained from the pomegranate leaf hydroalcoholic extract (PgEA), and identified a particularly interesting compound in it, galloyl-hexahydroxydiphenoil-glucose (G-HHDP-G), which is a hydrolyzable tannin whose pharmacological activities remain undetermined [18]. Later, Pinheiro et al. [19] showed that G-HHDP-G has anti-inflammatory properties and protects against acute lung injury in mice, and thus may be useful for the treatment of this condition and other inflammatory disorders. Nevertheless, there are no studies related to the antifungal effects of G-HHDP-G, and few have investigated the action of *P. granatum* leaf extract against *Candida* species. Therefore, we aimed to investigate the antifungal and antiviral actions of the PgEA fraction and G-HHDP-G against *Candida* spp. In addition, we also evaluated the antifungal activity of a combination of this fraction and compound with fluconazole (FCZ), to explore their synergistic interaction. This could help in the identification of compounds that can possibly serve as potential targets for the development of new herbal or drug formulations, in addition to providing a strategy for alternative therapies.

## 2. Results

### 2.1. In Silico Analysis of the Biological Activities of G-HHDP-G and Its Hepatotoxic Action

The biological activity spectra of G-HHDP-G were determined using an online version of Prediction of Activity Spectra for Substances (PASS) software. Table 1 shows the values obtained for the probable activity (Pa) and probable inactivity (Pi). Several activities were predicted for G-HHDP-G, including anti-infective, antioxidant, hepatoprotective, anti-inflammatory, immunostimulant, and antifungal. The highest Pa value was obtained for the anti-infective activity (0.962). The biological activities of FCZ were also determined for comparison. Table 2 shows the biological activity spectra of FCZ. The highest FCZ Pa value was for its antifungal activity (0.726).

The in silico predictions of chemical toxicity are shown in Table 3. The G-HHDP-G did not show any possible damage to the analyzed cytochromes, but FCZ showed probable hepatotoxicity.

### 2.2. Antifungal Activity

We evaluated the antifungal activity of PgEA and G-HHDP-G against a panel of two clinical and two reference strains of *Candida* spp. Both PgEA and G-HHDP-G exhibited antifungal activity against all the tested strains. However, the inhibitory concentrations varied among the isolates. The minimum inhibitory concentration (MIC) ranges were from 31.25 to 250 µg/mL for PgEA, from 31.25 to > 500 µg/mL for G-HHDP-G, and from 4 to 16 µg/mL for FCZ (Table 4).

### 2.3. In Vitro Interaction between PgEA/FCZ and G-HHDP-G/FCZ

The interactions between PgEA/FCZ and G-HHDP-G/FCZ were evaluated using a checkerboard assay. The fractional inhibitory concentration index (FICI) showed a synergistic interaction between PgEA/FCZ (Figure 1A–D) and G-HHDP-G/FCZ (Figure 1E,F) against different *Candida* species. According to the assay, the combination of PgEA and FCZ showed synergistic effects against all *Candida* isolates. On the other hand, the combination of G-HHDP-G/FCZ had a synergistic effect against two *Candida* strains (Table 5). Furthermore, there was a drastic reduction in the MIC values of compounds when they were used in combination, compared to the MICs obtained for each compound alone. For example, the MIC values for PgEA against *C. albicans* ATCC 90028 reduced from 125 µg/mL to 3.9 µg/mL when the fraction was used in combination with FCZ. Against *C. albicans* CAS, the reduction was from 250 µg/mL to 7.8 µg/mL. In case of G-HHDP-G, the values reduced from 125 µg/mL and 31.25 µg/mL to 31.2 µg/mL and 7.8 µg/mL against *C. glabrata* ATCC 2001 and *C. glabrata* FJF, respectively. Even the MIC values for FCZ reduced for most strains when it was used in combination the fraction and compound (Table 5).

### 2.4. Antibiofilm Effect

Figure 2 shows the effects of PgEA, G-HHDP-G, and FCZ on biofilm formation and preformed biofilms. All tested strains formed biofilms. PgEA and G-HHDP-G reduced the biofilm formation and significantly interfered with the preformed biofilms of both *C. albicans* and *C. glabrata* (*p* < 0.05), both at sub-inhibitory and higher concentrations (Figure 2 and Figure 3). FCZ was not able to interfere with the biofilm formation process of the strains, except at MIC concentrations. FCZ also did not interfere with the preformed biofilms of any strain. In contrast, all synergistic concentrations used were able to inhibit both stages of biofilm formation.

### 2.5. Time Kill-Curve Assay

To evaluate the time-kill activity, this assay was performed over a period of 36 h with *C. glabrata* ATCC 2001, and *C. albicans* ATCC 90028 in the presence of G-HHDP-G. The G-HHDP-G concentration of 2 × MIC and 3 × MIC can inhibit *C. glabrata* ATCC 2001 cell viability from 24–30 h when compared to 1 × MIC and the negative control (4A). Interestingly, for *C. albicans* ATCC 90028, 3 × MIC (2000 µg/mL) was responsible for completely eliminating viable cells within 12 h of exposure (_6_Log of cells/mL) (Figure 4B). The killing activity of G-HHDP-G appears to be dependent on the yeast species or strain; in turn, there is a degree of time and concentration dependence for microbial inhibition.

### 2.6. Phospholipase Assay

We measured the extracellular phospholipase activity of *C. albicans* ATCC 90028, *C. albicans* CAS, *C. glabrata* ATCC 2001, and *C. glabrata* FJF. *C. glabrata* FJF did not produce phospholipases. The higher the phospholipase activity (as measured according to the calculated phospholipase precipitation zone (Pz)), the lower the Pz value. Thus, both PgEA and G-HHDP-G significantly reduced phospholipase production (*p* < 0.05), by interfering with the enzyme production levels (Figure 5 and Table 6).

## 3. Discussion

In the present study, we report the antifungal activities of PgEA and one of its phenolic compounds G-HHDP-G. *P. granatum* has attracted the interest of researchers due to its main biological activities, including antioxidant, anti-inflammatory, antibacterial, anticancer, and antiviral [17,18,19,20,21]. Phytochemical analysis performed previously by our research group showed that a richness of phytochemical compounds is present in the PgEA fraction, which corroborates its biological properties [18,19,20]. Among these compounds, we highlight G-HHDP-G in the present study.

In silico analysis of G-HHDP-G indicated a potential antifungal effect of the compound, with a Pa value of 0.692. Pa and Pi values range from 0.000 to 1.000. When Pa is greater than Pi, the compound is believed to be experimentally active. Pa values ranged from 0.5 to 0.7, indicating that the compound will likely show considerable pharmacological effects experimentally [22], which corroborates the data obtained herein. We also evaluated the in silico effects and toxicity of G-HHDP-G and compared the results obtained with those for FCZ, which is the drug of choice for the treatment of fungal infections. The results highlighted low hepatotoxicity of G-HHDP-G in comparison to that of FCZ. Furthermore, the analysis showed that the compound has a potential hepatoprotective effect (Pa = 0.883), which was not observed in case of FCZ. The examinations were based on the structure-activity ratio of approximately 200,000 compounds and 4000 types of pharmacological activities [23].

Some polyphenolic molecules derived from gallic acid have recently been studied to understand their biological properties. Zhang et al. [24] reviewed 1,2,3,4,6-penta-O-galloyl-β-D-glucose, a gallotanin derivative, and drew attention to its attractive pharmacological and physiological activities, such as anticancer, apoptosis-inducing, anti-inflammatory, and antioxidative. Antiviral, antibacterial, and antibiofilm activities have also been attributed to some galotannin derivatives [25,26]. Similarly, Al-Sayed and Esmat [22] verified the hepatoprotective and antioxidant effects of pentagalloyl glucose and other galloyl esters isolated from the extract of *Melaleuca styphelioides*. However, studies on the antifungal effect of PgEA are scarce, with no reports in the literature about the potential of G-HHDP-G against *Candida*. Therefore, we decided to demonstrate the promising antifungal activity of PgEA and G-HHDP-G in vitro.

We confirmed the results obtained in silico by estimating the MIC. MIC values for PgEA ranged from 31.25 µg/mL to 250 µg/mL, which was effective against all the tested strains. However, G-HHDP-G was only effective against *C. glabrata* strains, with MIC values ranging from 31.25 µg/mL to 125 µg/mL. These results are extremely relevant because *C. glabrata* is intrinsically resistant to azoles [26]. The highest G-HHDP-G value tested against *C. albicans* was 500 µg/mL, and this concentration was not growth inhibitory. One possibility is that G-HHDP-G has an antifungal effect against *C. albicans* when combined with one of the other compounds present in PgEA, since this fraction inhibited the growth of this species.

Most of the studies related to the antifungal effects of *P. granatum* refer to extracts from the fruit, bark, or peel. Lavaee et al. [17] verified that the methanolic and ethanolic extracts of the bark and root of *P. granatum* had anti-*Candida* activity. *P. granatum* peel ethanol extract also showed antifungal activity against oral *Candida* isolates when tested using the agar well diffusion method [27], and there are few reports on the activities of the leaf extract against *Candida*. In a recent study [28], the authors found that after fractionation of the hydroalcoholic extract of *P. granatum* leaves, the ethyl acetate fraction was the richest in polyphenols. However, this fraction did not inhibit the growth of *C. albicans,* which differs from the results obtained in the present study.

To the best of our knowledge, there are no studies in the scientific literature that verify the antifungal activity of G-HHDP-G from PgEA. It is well known that *P. granatum* extracts have antifungal activity; however, the compounds responsible for this effect have not yet been identified, and whether they have antivirulence activity is still poorly understood. An interesting investigation was conducted by Brighenti et al. [29], who verified the effect of phenolic compounds from *P. granatum*, such as punicalin, punicalagin, ellagic acid, and gallic acid, on clinical and reference strains of *C. albicans* and found punicalagin to be the most active. Other studies have also identified punicalagin as the bioactive compound responsible for the antimicrobial activity of pomegranate peel [30,31].

In this study, both PgEA and G-HHDP-G inhibited most *Candida* strains at very low MIC values. However, combination therapies are increasingly being used in clinical trials, with the aim of decreasing conventional antimycotic side effects or toxicity and selection of resistant isolates [32]. Therefore, we decided to evaluate whether PgEA and G-HHDP-G, in combination with FCZ, would present a synergistic interaction against *Candida* strains. The FICI values obtained were very low, and the combinations displayed increased antifungal efficacy against *C. albicans* and *C. glabrata*, over the compounds alone. The MIC values of all the compounds against the tested strains reduced drastically by 50%–97% and 75%–88% for PgEA and G-HHDP-G, respectively, when used in combination. Synergistic interactions overcome the limitations of traditional antifungals by reducing the associated side effects and increasing their spectrum of action [33].

Endo et al. [34], when evaluating the synergistic effect of *P. granatum* fruit extract + FCZ against *C. albicans* isolates, verified that the MIC values for FCZ decreased two-fold when combined with the fruit extract. Similar results were obtained by Silva et al. [35], who evaluated the effects of the combinations of nystatin and punicalagin against *C. albicans*. Combined concentrations increased the antifungal efficacy as compared to the compounds alone, and the two of them reduced punicalagin’s MIC-50 by four- and eight-fold, increasing *Candida* inhibition and abrogating the cytotoxicity of punicalagin. These findings support our results since the MIC values for PgEA, G-HHDP-G, and FCZ decreased when they were combined with each other.

An alternative therapy for fungal infection treatment could be the use of compounds with action against. *Candida* spp. possesses several virulence factors that contribute to its high pathogenicity, including the ability to form biofilms. *Candida* species are capable of forming biofilms on both biotic (such as plant/animal cells and tissues) and abiotic surfaces (catheters, prosthetic devices, and dentures) [36]. Biofilms protect against cellular phagocytosis and make *Candida* cells resistant to antifungal drugs [37,38,39]. In our study, we evaluated the effects of PgEA, G-HHDP-G, and FCZ on biofilm formation and preformed biofilms.

PgEA and G-HHDP-G inhibited biofilm formation and reduced preformed biofilms of both *C. albicans* and *C. glabrata*, showing greater effectiveness than FCZ. In addition, the synergistic combinations of PgEA/FCZ and G-HHDP-G/FCZ were more efficient against *C. albicans* and *C. glabrata* biofilms than the substances alone. These results are extremely relevant because studies involving the effect of *P. granatum* on *Candida* biofilms are rare, and it is difficult to find effective compounds that efficiently inhibit biofilms. In a similar study, Bakkiyaraj et al. [40] also observed an antibiofilm action of the methanolic extract of *P. granatum* and its major compound ellagic acid, but at higher concentrations than those identified in our study. Almeida et al. [41] reported the antibiofilm activity of enriched fractions of *Equisetum giganteum* and *P. granatum* associated with and incorporated in a denture adhesive against *C. albicans*. The mixture was effective against the formation of biofilms on the surface of previously treated polymerized acrylic resin specimens. Villis et al. [42] used the same PgEA fraction as that in this study and verified that this fraction reduced the pre-formed biofilm of some *Cryptococcus* isolates, while showing better activity than FCZ. We highlight the importance of our findings in significantly reducing *Candida* biofilms, because these structures are generally associated with the majority of *Candida* infections and treatment failures, due to their drug-resistant biostructure [43,44,45]. Furthermore, this is the first study to assess the ability of PgEA, G-HHDP-G, and their combinations with FCZ to inhibit *Candida* biofilms.

Phospholipases are also relevant virulence factors produced by *Candida* spp. These are enzymes capable of breaking the phospholipid membranes or destroying proteins of the host immune system, and therefore, serve as relevant targets for antivirulence therapies [46]. In general, PgEA and G-HHDP-G significantly reduced phospholipase production (*p* < 0.5), as compared to FCZ, by interfering with the enzyme production levels. Liu et al. [32] showed that use of licofelone in combination with FCZ decreased the phospholipase activity at low concentrations, as compared to FCZ alone, with the inhibitory effect being positively correlated with the drug concentration. In turn, Nciki et al. [47] tested the effect of tannin-rich extracts in reducing phospholipase production in *Candida*, which required concentrations that were up to three times higher than those used in this study, suggesting that PgEA and G-HHDP-G have high antivirulence activity.

A limitation of our study is that the action of G-HHDP-G, alone and in association with fluconazole, was evaluated only against species of *C. albicans* and *C. glabrata*, since this was our main objective. Thus, there is a need to investigate a greater number of isolates of clinical origin from different anatomical sites and from different species in order to have a broader assessment of the findings of the present study. Additionally, we intend to carry out an evaluation of the effectiveness of the association of the compounds in experimental animal models.

## 4. Materials and Methods

### 4.1. Preparation of PgEA and Isolation/Identification of G-HHDP-G

The hydroalcoholic extract and ethyl acetate fraction of *P. granatum* leaves were obtained as described by Marques et al. [48] and Pinheiro et al. [18]. The ethyl acetate fraction was subjected to a silica gel chromatography column (230–400 mesh; 8 × 100 cm) and eluted with increasing polarities of mixtures of n-hexane/ethyl acetate and ethyl acetate/methanol, to obtain subfractions. The chromatographic separation resulted in 660 fractions. These fractions were grouped into 6 groups according to the similarity of the chromatographic profile. Group 6 was subjected to another round of chromatography and the compound was isolated in galloyl-HHDP-glucose of > 95% purity.

For Group 1, the following polarity gradient was used: Hexane (40%), Ethyl Acetate (60%) and Methanol (0%); Group 2, the following polarity gradient was used: Hexane (30%, 20% and 10%), Ethyl acetate (70%, 80% and 90%) 0% methanol; Group 3, the following polarity gradient was used: Hexane (0%), Ethyl Acetate (100% and 90%), and methanol (0% and 10%); Group 4, the following polarity gradient was used: Hexane (0%), Ethyl Acetate (80%) and Methanol (20%); Group 5, the following polarity gradient was used: Hexane (0%), Ethyl Acetate (70% and 60%) and Methanol (30% and 40%); Group 6 which contained galloyl-HHDP-glucose, the following polarity gradient was used: Hexane (0%), Ethyl Acetate (40%, 20% and 0%) and Methanol (60%, 80% and 100%). 

The structure was determined using HPLC-DAD-ESI-IT/MS analysis, as previously described by Pinheiro et al. [18] (Figure 6). The compound G-HDP-G was characterized with data obtained by fragmentation by mass spectrometry and compared with an authentic standard.

### 4.2. In Silico Analysis

#### 4.2.1. Prediction of the Biological Activities of G-HHDP-G In Silico

The biological activities of G-HHDP-G and FCZ (standard drug) were evaluated using PASS Online [version 2.0, Way2Drug.com©2011–2022, Moscow, Russia] (www.way2drug.com/passonline/, accessed on 16 September 2021), which provides several characteristics of the biological action of a substance. The PASS program describes biological activity as “active” (Pa) or “inactive” (Pi), in which the estimated probability varies from zero to one. The chances of finding a particular activity increase when the Pa values are higher and Pi values are lower. The results of PASS prediction were interpreted as follows: (i) only biological activities with Pa > Pi were considered possible for a particular compound; (ii) if Pa > 0.7, the substance is likely to exhibit biological activity and the probability of the compound being an analog of a known pharmaceutical drug is also high; (iii) if 0.5 < Pa < 0.7, the compound is likely to present biological activity, but the substance is not similar to known drugs; (iv) if Pa < 0.5, the chance of finding a biological activity is lower, but the chance to find a structurally new compound is greater.

#### 4.2.2. In Silico Analysis of G-HHDP-G Hepatic Toxicity

To assess the hepatic toxicity of G-HHDP-G and FCZ, we used the Super-CYPsPred [©Structural Bioinformatics Group 2019, Berlin, Germany] (http://insilico-cyp.charite.de/SuperCYPsPred/, accessed on 16 September 2021) web server, which includes machine learning models based on the random forest algorithm and different types of data sampling methods. The models presented in SuperCYSPred discriminate between inhibitors and non-inhibitors for the five main CYP450 isoforms. Fragment-based and structural similarity approaches were used to evaluate the applicability domain of the models, in addition to predicting a specific compound as active (inhibitor) or inactive (non-inhibitor) for a defined CYP isoform.

### 4.3. In Vitro Analysis

#### 4.3.1. Candida Strains and Growth Conditions

For the in vitro assays, we used two clinical isolates from vaginal samples (*C. albicans* CAS and *C. glabrata* FJF 2001; CEP/UNICEUMA no.: 813.402/2014) and two reference strains from the American Type Culture Collection (ATCC; *C. albicans* ATCC 90028 and *C. glabrata* ATCC 2001). The reference strains were kindly donated by the São Paulo State University, Araraquara Dental School, São Paulo, Brazil. Strains were plated on Sabouraud dextrose agar (SDA, Merck, Darmstadt, Germany), incubated for 48 h at 37 °C, and maintained on SDA during the experiments.

#### 4.3.2. MIC Determination

The MIC was determined using the broth dilution method, following the recommendations of the Clinical and Laboratory Standards Institute [49]. PgEA, G-HHDP-G, and FCZ solutions were diluted in RPMI-1640 (Sigma-Aldrich^®^, St. Louis, MO, USA) (pH 7.0) buffered with 0.165 M morpholinepropanesulfonic acid (MOPS; Sigma-Aldrich, St. Louis, MO, USA). Each substance was added to the first well of 96-well microplates (100 μL/well), with serial dilutions carried out in subsequent wells. The obtained and tested concentrations were 1000–1.95 μg/mL (PgEA), 500–0.07 μg/mL (G-HHDP-G), and 64–0.125 μg/mL (FCZ). Following that, 100 μL of *Candida* inoculum (1 × 10^3^ CFU/mL) was added to each well and incubated at 37 °C for 48 h in RPMI-1640 medium. After the incubation period, MIC was defined as the lowest concentration that visibly inhibited fungal growth. FCZ was used as a positive control and RPMI (100 µL) plus standardized inoculum was used as a negative control. The results were obtained from three independent assays performed in triplicate. 

#### 4.3.3. In Vitro Interaction Assays between PgEA + FCZ and G-HHDP-G + FCZ

Interactions between PgEA/FCZ and G-HHDP-G/FCZ were evaluated using the checkerboard test [49]. The following concentrations of PgEA, G-HHDP-G, and FCZ were used for each *Candida* strain: Combination 1: PgEA (250–0.97 μg/mL) and FCZ (16–0.06 μg/mL), for *C. albicans* ATCC 90028; Combination 2: PgEA (500–1.95 μg/mL) and FCZ (16–0.06 μg/mL), for *C. albicans* CAS; Combination 3: PgEA (62.5–0.24 μg/mL) and FCZ (32–0.125 μg/mL), for *C. glabrata* ATCC 2001; Combination 4: PgEA (62.5–0.24 μg/mL) and FCZ (8–0.03 μg/mL), for *C. glabrata* FJF. All the substances were diluted in RPMI-1640/MOPS medium.

One hundred microliters of the inoculum (1 × 10^3^ CFU/mL), 50 μL of PgEA or G-HHDP-G, and 50 μL of FCZ were added to 96-well plates. For sterility control, RPMI was used alone (100 µL), and growth was observed in RPMI (100 µL) plus standardized inoculum. Antimicrobial activity was assessed as described for MIC. After data normalization, the FICI was calculated for each compound, according to the general formula: FICI = [MICFCZ in combination/MICFCZ] + [MICPgEA in combination/MICPgEA] or FICI = [MICFCZ in combination/MICFCZ] + [MICG-HHDP-G in combination/MICG-HHDP-G]. FICI was calculated for all possible combinations of different concentrations against the same strain, and the final result was expressed as the mean of the FICI values. In addition, interaction curves were also constructed. The interaction between compounds was classified as synergism if FICI ≤ 0.5, indifferent if 0.5 > FICI ≤ 4.0, and antagonism if FICI > 4.0 [50]. Three independent assays were performed in triplicate.

#### 4.3.4. Effect of PgEA and G-HHDP-G on Candida Biofilms

*Candida* biofilms were developed using a slightly modified method [51,52]. To verify the interference of substances on biofilm formation, 200 μL of each substance at MIC, sub-inhibitory concentrations of MIC/4 and MIC/2, and established synergistic concentrations was used. The interference of substances on preformed biofilms was analyzed using the concentrations of MIC, 4× MIC, 8× MIC, and synergistic concentrations. PgEA and FCZ were tested against *C. albicans* (ATCC 90028 and CAS) and *C. glabrata* (ATCC 2001 and FJF). G-HHDP-G was tested against *C. glabrata* (ATCC 2001 and FJF). 

*Candida* cells previously grown in SDA were transferred to Yeast Nitrogen Base Broth (YNB) (Sigma-Aldrich^®^, St. Louis, MO, USA) and incubated for 18 h at 37 °C. The cell pellet was washed three times with sterile phosphate-buffered saline (PBS). A standard cell suspension (1 × 10^6^ CFU/mL, 200 μL) was added to 96-well plates and allowed to adhere for 90 min. After the adhesion phase, the microplates were gently washed three times with PBS to remove planktonic cells. To evaluate the interference on biofilm development, 200 μL of PgEA, G-HHDP-G, or FCZ, diluted in YNB + 100 mM glucose, were added to the corresponding wells, and the microplates were incubated for 24 h. For analysis of the interference on preformed biofilm, after the adhesion period, the wells were washed and each well was replaced with 200 μL of YNB. The microplates were then incubated for 24 h. Later, the wells were washed three times, 200 μL of PgEA, G-HHDP-G, and FCZ was added to the wells, and biofilms were incubated for a further 24 h. In all the experiments, biofilms without substances were used as controls.

After the final incubation, biofilms were evaluated for cell viability using 3-(4,5-dimethylthiazol-2-yl)-2,5-diphenyl-2H-tetrazolium bromide (MTT; Sigma-Aldrich) method [53]. Briefly, biofilms were washed with PBS, and 100 mL of MTT (5 mg/mL) was added to each sample and incubated for 4 h under light. Supernatants were then removed, 100 mL of Dimethyl Sulfoxide(DMSO) was added to each well, and the samples were incubated for another ten minutes. Readings were performed using a microplate reader (Softmax^®^ Pro-Molecular Devices General Counsel, USA) at the wavelength of 490 nm. Each experiment was conducted three times in triplicate.

#### 4.3.5. Time Kill-Curve Assay

The time-curve experiments were carried out in plastic tubes with screw caps in RPMI medium (Sigma-Aldrich), with a final volume of 500 µL at 37 °C for 36 h. The cells to the start of the experiment to obtain fungal cultures in early logarithmic phase growth. Cells were suspended in sterile distilled water to achieve a starting inoculum size of 1–5 × 10^6^ colony forming units (CFU)/mL and added to the tubes containing G-HHDP-G at concentrations 0.5, 1, 2, and 3 times the MIC. Growth control was also measured by adding the inoculum to tubes containing RPMI medium without drug. Sample for viable counts was taken at 0, 6, 12, 24, 30, and 36 h, plated in triplicate onto Sabouraud dextrose agar (SDA, Difco), and incubated for 24–48 h at 37 °C. After, incubation samples were first diluted in sterile saline (NaCl, 0.9%) and plated in the culture medium. Experiments were performed in duplicate for each isolate at different times. The results of the counts of the yeasts *C. albicans* ATCC 90028 *C. glabrata* ATCC 2001 were expressed in Log_10_ CFU/mL.

#### 4.3.6. PgEA, G-HHDP-G, and FCZ Interference in Phospholipase Production

The phospholipase activity of *Candida* spp. was determined using egg yolk agar medium. Both *C. albicans* and *C. glabrata* cultures (1 × 10^3^ CFU/mL) were treated with PgEA, G-HHDP-G, and FCZ at MIC, MIC/2, and MIC/4. A control group without substances was also included. The cultures were transferred into separate microtubes and incubated at 37 °C for 4 h. Subsequently, 10 μL of the suspension from each tube was inoculated into egg yolk agar medium and the plates were incubated at 37 °C for 72 h. After that, the diameters of the precipitation zones (a) and diameter of the precipitation zone plus diameter of the colony (b) were measured. The Pz was designated as Pz = a/b, as described by Price et al. [54] and Liu et al. [32]. According to this definition, the phospholipase production index was scored and categorized as follows: negative (Pz = 1), very low (Pz = 0.90 to 0.99), low (Pz = 0.80 to 0.89), high (Pz = 0.70 to 0.79), and very high (Pz ≤ 0.69) [50]. Each experiment was conducted three times in triplicate.

### 4.4. Statistical Analysis

All experiments were performed in triplicate, and the values have been expressed as mean ± standard deviation. The results were analyzed using one-way ANOVA, followed by Tukey’s test. Statistical analyses were performed using Prism 7.00 software (GraphPad, San Diego, CA, USA), and differences were considered significant when *p* < 0.05.

## 5. Conclusions

The present study provides a substantial advance over recent studies on *P. granatum* and its compounds, and, to the best of our knowledge, is the first to discover the antifungal effects of G-HHDP-G. We are also pioneers in verifying the synergistic effect of PgEA and G-HHDP-G, in combination with FCZ, against *Candida* spp. planktonic cells and biofilms. These results indicate that both the PgEA fraction and the compound G-HHDP-G are potential candidates that could serve as antifungal agents and promising synergists with FCZ for the development of new drugs against *Candida*. However, more in-depth studies need to be conducted to uncover the mechanisms of action of these compounds.

## Figures and Tables

**Figure 1 antibiotics-11-00265-f001:**
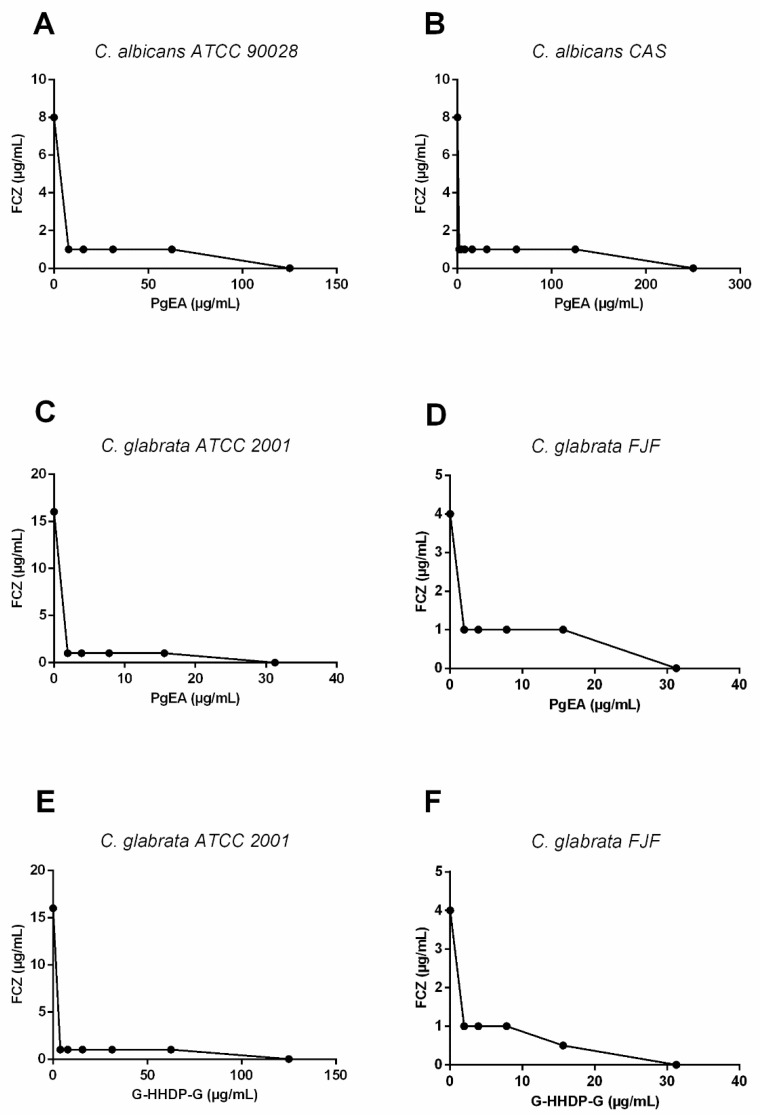
Interaction curves constructed for each pair of compounds when used in combination. Synergistic effect ethyl acetate fraction of *P. granatum* (PgEA) and fluconazole FCZ (**A**–**D**) against *C. albicans* (**A**,**B**) and *C. glabrata* (**C**,**D**) strains. Synergistic effect of galloyl-hexahydroxidifenoil-glucose (G-HHDP-G) and FCZ against *C. glabrata* strains (**E**,**F**).

**Figure 2 antibiotics-11-00265-f002:**
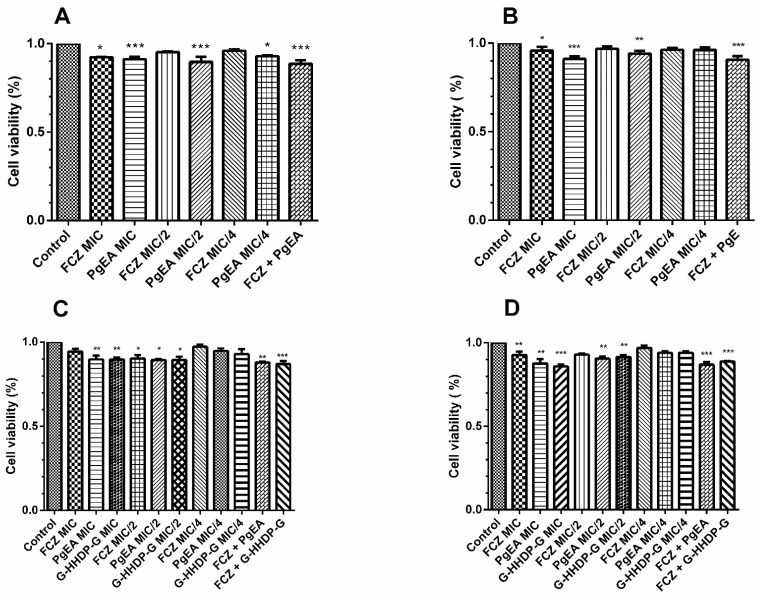
Effect of the ethyl acetate fraction of *P. granatum* (PgEA), galloyl-HHDP-glucose (G-HHDP-G), and Fluconazole (FCZ) on *Candida* biofilm formation. (**A**) Effect of PgEA and FCZ on the biofilm formation of the reference strain *C. albicans* 90028, using inhibitory, sub-inhibitory, and synergistic concentrations. (**B**) Effect of PgEA and FCZ on biofilm formation of the clinical isolate *C. albicans* CAS, using inhibitory, sub-inhibitory, and synergistic concentrations. (**C**) Effect of PgEA, G-HHDP-G, and FCZ on biofilm formation of the reference strain *C. glabrata* 2001, using inhibitory, sub-inhibitory, and synergistic concentrations. (**D**) Effect of PgEA, G-HHDP-G, and FCZ on the biofilm formation of the clinical isolate *C. glabrata* FJF, using inhibitory, sub-inhibitory, and synergistic concentrations. * *p* < 0.05, ** *p* < 0.01, and *** *p* < 0.0001. Data represent the mean ± SD of three independent experiments carried out in triplicate.

**Figure 3 antibiotics-11-00265-f003:**
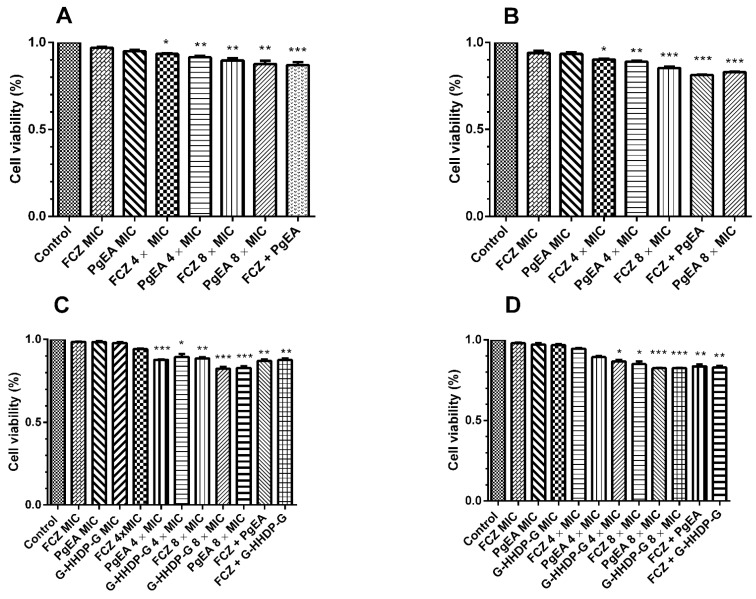
Effect of the ethyl acetate fraction of *P. granatum* (PgEA), galloyl-HHDP-glucose (G-HHDP-G), and Fluconazole (FCZ) on preformed biofilms of *Candida* spp. (**A**) Effect of PgEA and FCZ on the preformed biofilm of the reference strain *C. albicans* 90028, using inhibitory, higher than inhibitory, and synergistic concentrations. (**B**) Effect of PgEA and FCZ on the preformed biofilm of the isolate of clinical *C. albicans* CAS, using inhibitory, higher than inhibitory, and synergistic concentrations. (**C**) Effect of PgEA, G-HHDP-G, and FCZ on the preformed biofilm of the reference strain *C. glabrata* 2001, using inhibitory, higher than inhibitory, and synergistic concentrations. (**D**) Effect of PgEA, G-HHDP-G, and FCZ on the preformed biofilm of the clinical isolate *C. glabrata* FJF, using inhibitory, higher than inhibitory, and synergistic concentrations. * *p* < 0.05, ** *p* < 0.01, and *** *p* < 0.0001. Data represent the mean ± SD of three independent experiments carried out in triplicate.

**Figure 4 antibiotics-11-00265-f004:**
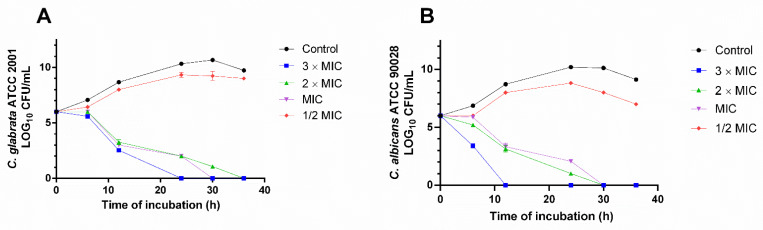
Time–kill curve for *C. glabrata* ATCC 2001 (**A**) and *C. albicans* ATCC 90028 (**B**). G-HHDP- G compound was tested at 3 × MIC. MIC: minimal inhibitory concentration. CFU: colony-forming unit. Time is expressed in hours. Negative control: no compound was added to the cell suspension.

**Figure 5 antibiotics-11-00265-f005:**
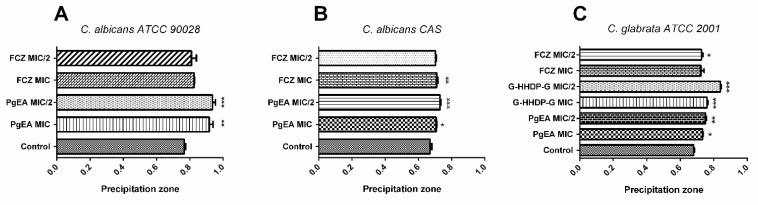
Evaluation of the degree of interference mediated by effect ethyl acetate fraction of *P. granatum* (PgEA), Galloyl-Hexahydroxidifenoil-Glucose (G-HHDP-G), and Fluconazole (FCZ) on *Candida* spp. phospholipase production. (**A**) *C. albicans* ATCC 90028, (**B**) *C. albicans* CAS, and (**C**) *C. glabrata* ATCC 2001. * *p* < 0.05, ** *p* < 0.01, and *** *p* < 0.0001.

**Figure 6 antibiotics-11-00265-f006:**
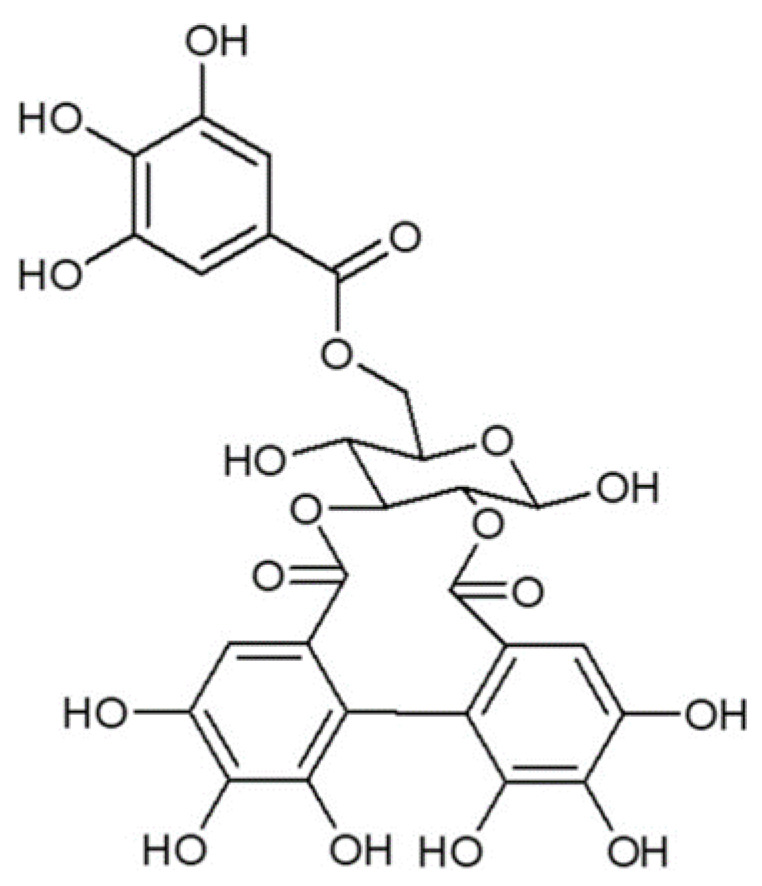
Chemical structure of Galloyl-Hexahydroxidifenoil-Glucose (G-HHDP-G), isolated from the ethyl acetate fraction of the *P. granatum* crude extract.

**Table 1 antibiotics-11-00265-t001:** In silico analysis of the biological activities of G-HHDP-G.

Activities	PASS Predictions of G-HHDP-G	
	Pa	Pi
Anti-infective	0.962	0.003
Antioxidant	0.895	0.003
Hepatoprotective	0.883	0.002
Anti-inflammatory	0.775	0.008
Immunostimulant	0.752	0.011
Antifungal	0.692	0.015

Pa, probable activity; Pi, probable inactivity; PASS, prediction of activity spectra for substances; G-HHDP-G, galloyl-hexahydroxidifenoil-glucose.

**Table 2 antibiotics-11-00265-t002:** In silico analysis of the biological activities of Fluconazole.

Activities	PASS Predictions of FCZ	
	Pa	Pi
Antifungal	0.726	0.008
ATPase inhibitor of phospholipid translocation	0.480	0.069
Cell wall synthesis inhibitor	0.351	0.002
NADPH inhibitor-cytochrome-c2 reductase	0.366	0.134

Pa, probable activity; Pi, probable inactivity; PASS, prediction of activity spectra for substances; FCZ, fluconazole.

**Table 3 antibiotics-11-00265-t003:** In silico prediction of chemical toxicity in hepatic cytochromes.

Cytochrome	Predicted Values of Inhibitory Effect	
	G-HHDP-G	FCZ
CYP1A2	NT (0.8) *	T (0.606) **
CYP2C19	NT (0.872) *	NT (0.775) *
CYP2C9	NT (0.796) *	T (0.698) **
CYP2D6	NT (0.734) *	T (0.502) **
CYP3A4	NT (0.704) *	T (0.572) **

NT, non-toxic; T, toxic; * 0.7–0.9, no expected toxicity; ** 0.5–0.7, predicted toxicity.

**Table 4 antibiotics-11-00265-t004:** MICs of PgEA, G-HHDP-G, and FCZ against *Candida* spp. The readings of cell turbidity were recorded after a 48 h incubation, at 37 °C, in RPMI-1640 medium.

Strains	MIC (µg/mL)		
	PgEA	G-HHDP-G	FCZ
*C. albicans* ATCC 90028	125 ± 0	>500 ± 0	8 ± 0
*C. albicans* CAS	250 ± 0	>500 ± 0	8 ± 0
*C. glabrata* ATCC 2001	31.25 ± 0	125 ± 0	16 ± 0
*C. glabrata* FJF	31.25 ± 0	31.25 ± 0	4 ± 0

The assays were performed in triplicate. MIC, minimal inhibitory concentration; PgEA, ethyl acetate fraction of *P. granatum*; G-HHDP-G, galloyl-hexahydroxidifenoil-glucose; FCZ, fluconazole.

**Table 5 antibiotics-11-00265-t005:** FICI and classification of the interaction between PgEA or G-HHDP-G and FCZ. The interaction was classified as synergism if FICI ≤ 0.5, non-interaction if 0.5 > FICI ≤ 4.0, and antagonism if FICI > 4.0.

Strain	MIC in Combination (µg/mL)			FICI			
	FCZ	PgEA	G-HHDP-G	FICI_FCZ+PgEA_	It	FICI_FCZ+G-HHDP-G_	It
*C. albicans* ATCC 90028	8	3.9	-	0.32	SYN	-	-
*C. albicans* CAS	1	7.8	-	0.36	SYN	-	-
*C. glabrata* ATCC 2001	4	15.6	31.2	0.45	SYN	0.47	SYN
*C. glabrata* FJF	0.5	7.8	7.8	0.49	SYN	0.37	SYN

MIC, minimum inhibitory concentration; FCZ, fluconazole; PgEA, ethyl acetate fraction; G-HHDP-G, galloyl-hexahydroxidifenoil-glucose; FICI, fraction inhibitory concentration index; It, interaction type; SYN, synergistic.

**Table 6 antibiotics-11-00265-t006:** *Candida* spp. extracellular phospholipase activity when treated with PgEA, G-HHDP-G, and FCZ, as evaluated in egg yolk medium in terms of the precipitation zone.

Treatments	Precipitation *Zone*			Phospholipase Activity
	*C. Albicans* *ATCC 90028*	*C. Albicans* *CAS*	*C. Glabrata* *ATCC 2001*	
Control	0.76	0.67	0.68	H/VH/VH
PgEA MIC	0.91	0.70	0.73	VL/H/H
PgEA MIC/2	0.93	0.73	0.75	VL/H/H
G-HHDP-G MIC	---	---	0.76	H
G-HHDP-G MIC/2	---	---	0.84	L
FCZ MIC	0.82	0.71	0.72	L/H/H
FCZ MIC/2	0.81	0.70	0.70	L/H/H

The precipitation zone represents the ratio of the diameter of the colony to the cloudy zone and colony diameter. VL: very low (Pz = 0.90 to 0.99); L: low (Pz = 0.80 to 0.89); H: high (Pz = 0.70 to 0.79); VH: very high (Pz ≤ 0.69). PgEA—Ethyl acetate fraction of *P. granatum*; G-HHDP-G—Galloyl-Hexahydroxidifenoil-Glucose; FCZ—Fluconazole.

## Data Availability

Not applicable.

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
