# Peer review of "Ethyl Acetate Fraction of Punica granatum and Its Galloyl-HHDP-Glucose Compound, Alone or in Combination with Fluconazole, Have Antifungal and Antivirulence Properties against Candida spp."

_antibiotics, 2022, doi:10.3390/antibiotics11020265_

Round 1

Reviewer 1 Report

Dear authors, this could be a proper article for publication. Some minor suggestions are the following:

1) I suggest to include information about fluconazole in the introduction section (describe it, some general lines about its use, why it was chosen to study its synergism...)

2) Results for the assay evaluating the antifungal activity (section 2.2) do not provide information about negative or positive controls. 

3) Section 4.1. Provide the gradient of polarities used for the chromatographic purification.

4) Could be possible to enlarge the font size of Figure 4, especially the axis?

Author Response

We thank the reviewers for all the expert critiques and constructive comments on the manuscript. We have revised it accordingly. Our point-by-point responses to the reviewers’ comments are in attached document. Changes that were made are highlighted in yellow in Microsoft Word in the revised manuscript.

Reviewer 2 Report

Interesting work, which fits well into the urgent search for new molecules with antimicrobial activity

The study is well structured from a microbiological point of view, and makes use of classic microbiology techniques,  and for this reason I only have one comment and one request:

  • To give more relevance to the study, especially with regard to the calculation of MIC values, perhaps the Authors could have analyzed a greater number of clinical strains, and also other clinically relevant species of the genus Candida.
  • As the Authors know, to confirm the results of the checkerboard studies, it is preferred to add time killing experiments, which I ask to be added at this stage of review

Finally, the word "analyses" must be corrected to "analysis" throughout the text.

Author Response

(The authors gave the same response as above.)

Reviewer 3 Report

Manuscript "Ethyl acetate fraction of Punica granatum and its galloyl-HHDP-glucose compound, alone or in combination with Fluconazole, have antifungal and antivirulence properties against Candida spp." presents the results of antifungal studies of natural substances and in combination with an antibiotic. Due to the increasing resistance of pathogens to popular antibiotics, new substances with antimicrobial properties are being sought. Manuscript is well written, research is well planned. The description of the research results is clear and the discussion is correct. The study of phospholipase activity as a factor influencing virulence is interesting and quite rarely studied, which makes the manuscript more interesting.

Detailed comments:

Introduction - you can supplement the information which natural substances have antifungal activity against C. albicans (extracts, oils, bee products, e.g. doi.org/10.3390/molecules24162965, doi.org/10.3390/antibiotics11010073).

Chapter 4.3.3. - In what medium was the inoculum prepared? 

Author Response

(The authors gave the same response as above.)

Reviewer 4 Report

This manuscript written by Aline Michelle S. Mendonça and co-authors described the antifungal and antivirulence properties of the ethyl acetate fraction of Punica granatum. Meanwhile, the bioactive of galloyl-HHDP-glucose alone or in combination with Fluconazole was evaluated. I think the experimental data are sound and accurate. I think the present manuscript can be accepted after minor revision.

 The comments are listed below:

  1. Line 28, In silicon…. This sentence seems not toc finish. The authors should also mention what kind of activity or delete this sentence.
  2. For the determination of G-HHDP-G, I think the authors should use 1H and 13C NMR spectra along with mass data.

Author Response

(The authors gave the same response as above.)

Round 2

Reviewer 2 Report

I thank the authors for having provided for what was requested.  The work in the revised form is now, in my opinion, publishable